# Machine learning based tuberculosis (ML-TB) health predictor model: early TB health disease prediction with ML models for prevention in developing countries



Priyanka Karmani[1], Aftab Ahmed Chandio[1], Imtiaz Ali Korejo[1], Oluwarotimi Williams Samuel[2] and Majed Aborokbah[3]

[1] Institute of Mathematics and Computer Science, University of Sindh, Jamshoro, Sindh, Pakistan
[2] School of Computing and Data Science Research Center, University of Derby, Derby, United Kingdom
[3] Faculty of Computers and Information Technology, University of Tabuk, Tabuk, Saudi Arabia

## ABSTRACT

**Background:** Tuberculosis (TB) remains one of the top infectious killers in the world and a prominent fatal disease in developing countries. This study proposes a prototypical solution to early prevention of TB based on its primary symptoms, signs, and risk factors, implemented by means of machine learning (ML) predictive algorithms. Further novelty of the study lies in the uniqueness of patient dataset collected from three top-ranked hospitals of Sindh, Pakistan, *via* a self-administered survey patient-records that comprises a set of questions asked by the doctors treating TB patients in real-time. A total of 1,200 survey patient-records were evenly distributed among all three hospitals, viz. ICT Kotri, LUMHS Jamshoro, and Civil Hospital Hyderabad.

**Methods:** To develop the required prototypes, the research made use of five distinct benchmark ML algorithms: decision tree (DT), Gaussian naive Bayes (GNB), logistic regression classifier (LRC), adaptive boosting (AdaBoost), and neural network (NN), whose performance was evaluated by considering various performance metrics, *i.e.*, accuracy, precision, recall, F1 score, and confusion matrix.

**Results:** The experimental results, graphically visualized and systematically discoursed, demonstrate that early detection of TB classifiers, including DT, GNB, LRC, AdaBoost, and NN, attained accuracy rates of 92.11%, 89.04%, 90.35%, 93.42%, and 92.98%, respectively. These results indicate effective diagnosis of TB disease by each implemented ML algorithm.

## INTRODUCTION

Among many toxic diseases, one of the leading fatal diseases is tuberculosis (TB), defined as, "an infectious disease caused by bacillus *Mycobacterium tuberculosis*". TB primarily communicates a disease to human respiratory tract *i.e.*, Lungs. According to World Health Organization (WHO), Pakistan ranks fifth among the 30 high TB-burden countries

Corresponding author
Aftab Ahmed Chandio,
chandio.aftab@usindh.edu.pk

globally (*The World Health Organization, 2024a*). To get control of this lethal disease and reduce death ratio, ensuring good health and well-being, it is essential to put forward an automated solution for TB diagnosis in its initial phase. In this contemporary era, the integration of cutting-edge technologies in the field of Medical and Health sciences, called Healthcare Informatics (HI), attempts to originate innovative and digital solutions as well as novel apparatuses to save human lives in an effective manner. Consequently, current research carried out a study on an automated machine learning (ML) oriented solution to predict about TB manifestation in an individual at an earlier stage.

The contribution of this research is four-fold: (a) the proposed systematic architecture of the ML-TB predictive model; (b) novel data collection from hospitals *via* self-administered TB screening questionnaire; (c) data analysis in Python to generate the targeted ML solutions using its primary signs, symptoms and risk factors; (d) the context of the research is Sindh, a high TB-burden province of Pakistan. Major highlights and insights found in the article are: (a) implemented ML algorithms have shown highly accurate results; (b) consideration of primary TB features for its automated diagnosis is vital for doctors to increase awareness and prevent the individuals from TB complications; and (c) the produced solution aids the doctors to diagnose and treat the patients at primary phase of TB that certainly reduces TB death ratio.

The rest of the article is organized as follows: In "Healthcare Informatics (HI) and Machine Learning (ML)", the Healthcare Informatics and machine learning are defined comprehensively. In "The Targeted Disease: Tuberculosis (TB)", the targeted disease tuberculosis is discussed. In "Literature Review", the extensive literature review is cited and the rationale of research problem is identified. "ML-TB Predictor" explains the methodology adopted in the research. "Data Analysis" shows the data analysis. "Results and Discussion" represents the experimental settings, results and discussions. Finally, "Conclusions" concludes the article and suggests future research directions.

## HEALTHCARE INFORMATICS AND MACHINE LEARNING

The U.S. National Library of Medicine described HI as "an interdisciplinary study of design, development, adoption, and implementation of modern technical apparatuses and processes in healthcare system" (*Parry, 2014*). Computer systems, clinical guiding principles, proper diagnostic taxonomies, information and communication systems are some of the resources used in HI. To manage patient healthcare, either individually or in groups, computational intelligence is used. In general, HI is motivated to advance the inclusive efficacy of patient healthcare (*Holzinger, 2016*). At present, machine learning (ML–a subdomain of Artificial Intelligence), is significantly integrated in the sphere of Healthcare Informatics (HI). The pioneer of computer gaming and Artificial Intelligence (AI), Arthur Samuel, in 1959 formulated the term ML, and described ML as the branch of research that enables computers to learn without being explicitly programmed (*The World Health Organization, 2024b*). To develop machines (algorithms) which are self-learner, self-checker, self-improviser and decision maker is the ultimate goal of ML. ML system learns from past data (*i.e.*, training), constructs a mathematical model, and provides predictions or decisions whenever it receives new data (*i.e.*, testing). The amount of data influences the accuracy of predicted output as shown in Fig. 1. ML combines computer

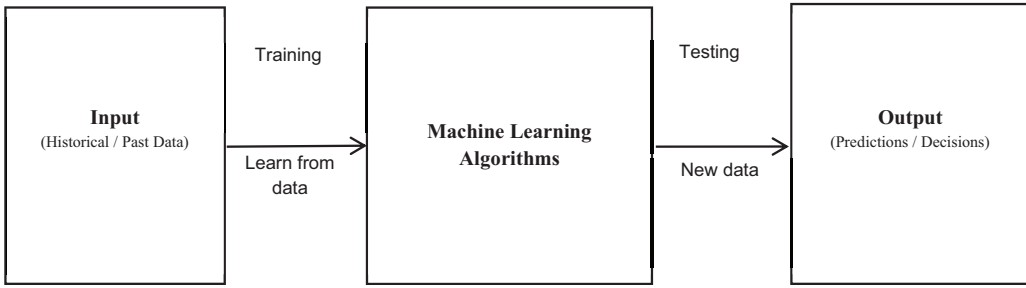

**Figure 1 Machine learning ML-TB predictor work flow.**

science and statistics to create predictive models; the former focuses on problem solving and deciding if problems are attainable throughout all phases, while the latter emphasizes on data modeling, reasoning, and reliability measurement.

The contribution of ML in HI is motivated towards predictive analytics, disease diagnosis, therapeutic formation, and other areas. For efficient and precise diagnosis, ML combines healthcare with cutting-edge computer technology, as well as assisting medical experts in providing superior care.

## THE TARGETED DISEASE: TUBERCULOSIS

As defined by *Kumar & Clark (2012)*, "TB is a communicable disease triggered by a microbes (bacilli) called *Mycobacterium tuberculosis* which usually blowout through air when an infected person cough, spit, speak, or sneeze". It primarily disrupts human respiratory tract (*i.e.*, the lungs). However, this bacterium can blowout and communicate the disease to other body parts. The former type is pulmonary TB while the latter is extra pulmonary TB. It may be either in one of two possible states *i.e.*, latent TB and active TB. In latent TB, the mycobacteria remains in a dormant state in the subject, however, in active TB the subject exhibits the relevant signs and symptoms. According to WHO, TB is one among the top 10 global fatal diseases. Pakistan ranks fifth highest TB-burden country and fourth highest prevalence of MDR-TB. In Pakistan, millions of people are infected with TB and 66% of the patients die every year. Since the turn of the century, the Pakistan National TB Program has made significant improvements for TB detection and treatment. In the beginning of previous decade, the rate of detection for TB cases raised from 19% to 84%. Within the same period, the success ratio of TB treatment raised to 91% (*The World Health Organization, 2024b*). Despite the statistics showing remarkable improvement, it is important to continue research on the treatment and diagnosis methods. Consequently, this research aims to improve the TB detection methods using state-of-the-art.

## LITERATURE REVIEW

In this section, an in-depth exploration of existing literature is conducted to provide a thorough understanding of the subject. Various ML types, specifically interactive machine learning (iML) and automated machine learning (aML), as well as ML approaches including Regular and Ensemble ML, are examined. Additionally, learning paradigms such as supervised, unsupervised, and semi-supervised, along with algorithms like decision tree,

support vector machines, naïve Bayes, regression analysis, neural networks, k-nearest neighbor, k-means clustering, genetic algorithms, deep learning, and Ensembles, have been explored in HI in our previous studies (*Karmani et al., 2018*, *2020*) and other studies (*Tasci, Uluturk & Ugur, 2021*; *Li et al., 2023*; *Zhang et al., 2023*; *Islam et al., 2022*). *Yahiaoui, Er & Yumusak (2017)* used the support vector machine (SVM) method to detect TB at the early stage, achieving 96.68% accuracy and low running costs. However, the study does not include a comparison of SVMs to other approaches to correctly determine its superiority. *Alcantara et al. (2017)* proposed a deep convolutional neural network model that incorporates mobile health technologies to improve TB detection in Peru, with accuracy rates of 89.6% for binary classification and 62.7% for multi classification. Nevertheless, the reliability and accessibility of mobile health technology may have an influence on the outcomes due to the variability of environmental settings. *Shahaboddin et al. (2014)* detected TB using hybrid ML techniques that included artificial immune systems (AISs), genetic algorithms, and neural networks, with 99.14% classification accuracy, 87% sensitivity, and 86.12% specificity. However, these hybrid ML systems can be computationally complex and intensive on resources, thus restricting their practical viability, particularly in resource-constrained contexts. *Er, Yumusak & Temurtas (2012)*, *(2010)* used AIS to identify multiple chest diseases, including TB, attaining 90% accuracy. In another study, they used multiple neural networks, including ANN, MLNN, PNN, LVQNN, GRNN, Bayesian networks, and RBFNN, and acquired 90% classification accuracy rate via MLNN. Yet, AIS and certain NN architectures could find it challenging to scale to more extensive datasets, limiting their performance and generalizability. Furthermore, owing to hyper parameter tuning, such techniques tend to be time-consuming and occasionally fail to yield flawless outcomes, particularly when dealing with limited computational resources (*Shahaboddin et al., 2014*).

As shown in Table 1, prior studies focused on TB disease diagnosis has taken into consideration complicated pathological features and no studies have been carried out in Pakistan, despite its high TB prevalence.

The rationale for undertaking this research in Pakistan, notably in Sindh province, arises from the region's high prevalence of TB as well as its peculiar epidemiological infrastructure, and socio-cultural challenges. TB epidemiology in Pakistan differs from that in other regions, requiring context-specific treatments to promote effective control. Inadequate healthcare infrastructure, along with cultural and socioeconomic dynamics, impedes TB detection and control initiatives, reinforcing the necessity for innovative methods tailored to the Pakistani context. Accordingly, present study aims to improve early detection and treatment initiation by developing an automated TB diagnostic model based on the analysis of its preliminary signs, symptoms, and risk factors. The research expects to reduce TB-related morbidity and mortality ratio in the country, contributing to positive health solutions for the affected populations.

## ML-TB PREDICTOR

According to the philosophical traits of research, the present study is a quantitative type of research study in general based on the positivist paradigm and adopted deductive

**Table 1 Literature review of ML-TB predictor.**

| Ref# | Dataset | Features | ML algorithm | Training procedure | Success ratio (%) |
|------|---------|----------|--------------|--------------------|-----|
| El-Solh et al. (1999) | 563 patients, State University of New York | 21 | GRNN (1 hidden layer) | - | 92.30 |
| dos Santos et al. (2007) | 136 patient's medical reports, University Hospital, Rio de Janeiro, Brazil | 26 | MLNN with BP (1 hidden layer) | | 77 |
| Er, Temurtas & Tanrikulu (2010) | 150 patient's medical report, Diyarbakir Chest Diseases Hospital, Turkey | 38 | GRNN (1 hidden layer) MLNN (1 hidden layer) MLNN (2 hidden layers) | BPwM LM | 93.18 93.04 93.24 93.93 95.08 |
| Elveren & Yumusak (2011) | 150 patient's medical report, Diyarbakir Chest Diseases Hospital, Turkey | 38 | MLNN (2 hidden layers) | GA | 94.88 |
| Dongardive et al. (2011) | 250 sample reports, T.B. Hospitals, Mumbai | 19 | Decision tree | IDT | 94.50 |
| Omisore, Samuel & Atajeromavwo (2017) | 10 TB patients from St. Francis Catholic Hospital, Delta State, Nigeria | 24 | MLNN | GA + NN + Fuzzy Logic | 70 |
| Lakhani & Sundaram (2017) | Montgomery County (MC), Shenzhen, Thomas Jefferson University Hospital and Belarus Tuberculosis Portal | CXRs | CNN | Ensemble Radiologist Augmented | 96.00 98.70 |
| Hwang et al. (2016) | Private Dataset, Montgomery County (MC) and Shenzhen Datasets | CXRs | CNN | Modified AlexNet | 90.00 |
| Lopes & Valiati (2017) | Montgomery County (MC), Shenzhen | CXRs | CNN | Ensemble | 84.60 |
| Alcantara et al. (2017) | 5,000 CXRs from Partners in Health at Peru and various other image DBs | CXRs | CNN | Binary and Multi-class Classification | 89.60 62.07 |

approach. According to nature, it is analytical research concerned with determining the cause-and-effect associations between two or more variables. According to the purpose of study, it is fundamental (basic or pure) research producing knowledge and understanding in relation to natural occurrences. The research design of the current study is conclusive (*i.e.*, decision making) as shown in Fig. 2. This research study adopted a low-cost, efficient, and precise survey method for data collection about a population, and it was administered to a large sample. The survey was designed and administered in accordance with the research guidelines of the University. Furthermore, this research study has been reviewed and approved by the Institutional Bioethics Committee (IBC) of the University (*i.e.*, Ref. No. ORIC/SU/311). The consent form has been received from the participants in written format which asks their agreement and the hospital details.

The main research instrument was a cross-sectional self-administered TB screening questionnaire consisting of four parts comprising personal information about the patient, exploratory questions relevant to TB disease, supplementary information, and comments/feedback. The questionnaire was quasi-structured, principally consisting of dichotomous (Yes/No) questions, along with two partial open-ended questions. In addition, to eliminate bias, the research made use of random probability sampling techniques. The questionnaire

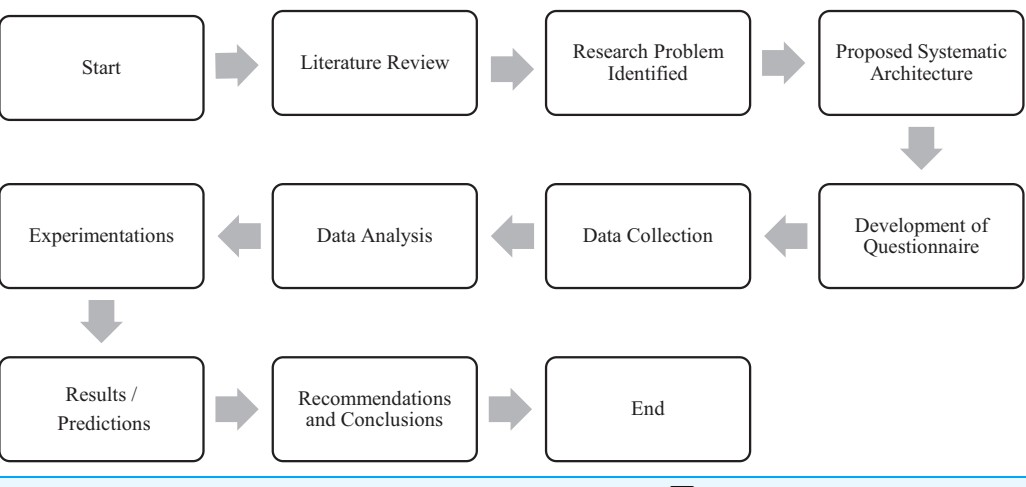

**Figure 2** Research design and steps.     

**Table 2 Statistical summary of the survey responses.**

| Hospitals | Number of questionnaires | |
|---|---|---|
| | Distributed | Returned |
| Institute of Chest Diseases (TB Sanatorium), Kotri | 400 | 360 |
| Liaquat University of Medical and Health Sciences, Jamshoro | 400 | 280 |
| Civil Hospital, Hyderabad | 400 | 310 |
| Aggregate | 1,200 | 950 |
| Ambiguous questionnaires | (−) 190 | |
| Final sample of the study | $n = 760$ | |

was pre-tested to ascertain its precision and psychometric reliability. Statistically, 1,200 questionnaires were distributed evenly across three existing hospitals in Sindh, namely the Institute of Chest Diseases (TB Sanatorium) Kotri, Liaquat University of Medical and Health Sciences (LUMHS) Jamshoro, and Civil Hospital Hyderabad. The questionnaires were administered by the researcher and answered by the doctors consulting TB-infected patients at these chosen hospitals, resulting in varied response rates. The highest response rate (90%) was obtained from the Institute of Chest Diseases (TB Sanatorium) Kotri, followed by 77.5% from Civil Hospital Hyderabad and 70% from LUMHS Jamshoro. In total, 950 responses were received, representing a response rate of 79.16%. After eliminating 190 responses (*i.e.*, missingness records) with partial data, 760 responses were chosen for further analysis as depicted in Table 2.

The aggregate response rate for the research was 63.3%. The summary provided in Table 3 shows that out of the 760 patients, 58.9% were male and 41.1% were female. The highest response rate was obtained from the 31–40 age bracket, while the lowest was in the 61–70 age bracket. The imbalance dataset across age groups was addressed *via* regulating the model with weighted parameters. Figure 3 illustrates the proposed systematic architecture, which is used to predict TB disease at an early stage *via* the implementation of ML algorithms to collected data sample.

**Table 3 Distribution of participants age group.**

| Age groups | ICD Kotri | | LUMHS Jamshoro | | Civil hospital Hyderabad | | Net ratio |
|---|---|---|---|---|---|---|---|
| | Male | Female | Male | Female | Male | Female | |
| 0–10 | – | – | – | – | – | – | – |
| 11–20 | 24 | 10 | 20 | 8 | 14 | 10 | 9.30% |
| 21–30 | 34 | 22 | 43 | 17 | 17 | 14 | 25.80% |
| 31–40 | 44 | 38 | 42 | 25 | 39 | 22 | 37.40% |
| 41–50 | 36 | 31 | 19 | 11 | 28 | 23 | 24.70% |
| 51–60 | 30 | 22 | 17 | 19 | 18 | 22 | 2.30% |
| 61–70 | 15 | 14 | 3 | 1 | 5 | 3 | 0.50% |
| Total | 183 | 137 | 144 | 81 | 121 | 94 | 100% |
| | 57.18% | 42.80% | 64% | 36% | 56.27% | 43.70% | |
| Grand total | 320 | | 225 | | 215 | | 760 |

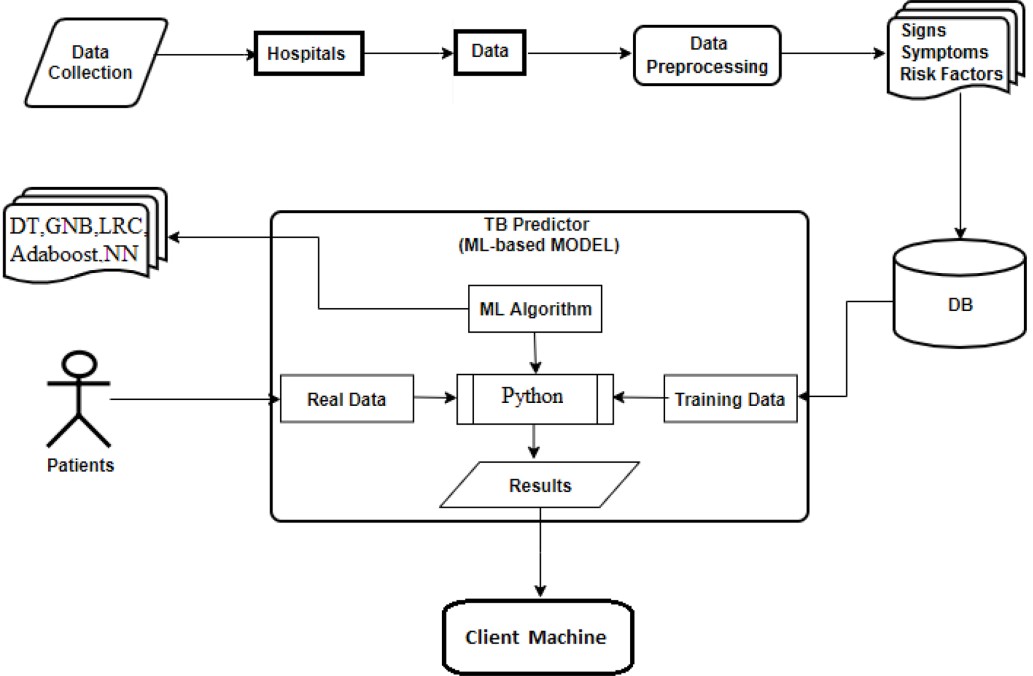

**Figure 3 The systematic architecture of the proposed ML-TB predictor.**

# DATA ANALYSIS

To create an ML model, the first crucial step is data preprocessing that prepares the raw data to make it feasible for the ML model as the real-time data usually encompasses noises, mislaid values, and impracticable format. Thus, data preprocessing is a significant procedure for cleaning and formatting the data in accordance with the required ML model. This practice also ensures the efficiency and accuracy of the ML model. Data analysis tools used in this research includes MS Excel, Jupyter Notebook and Python programming

language. Box 1 depicts the systematic data preprocessing for the current study in sequential order, including statistical information shown in Tables 4, 5 and visualizations in Figs. 4, 5. The preprocessed dataset comprises diverse features and their possible values (*i.e.*, as described in Table 5) including cough (varies in severity and frequency); cough duration (mild, occasional, intermittent, moderate, frequent or persistent); cough type (productive, or non-productive); mucus (clear, presence, or bloody color) aids to identify the type of infection; chest pain; breathe state (normal, or dyspnea); body temperature (fever or normal); chills; pulmonary effusion (presence of fluid in lungs); Erythrocyte Sedimentation Rate (ESR) value (distributed or normal); diet (nutritional status); physique (the healthy body structure or weight loss); energy adequacy (stamina level fatigue or fit); smoking; crowding (living in a cramped settings) and exposed (in direct contact with TB risk factors or pathogens).

---

**Box 1  The systematic data preprocessing.**

Following are the steps that describe the systematic data preprocessing for the current study in sequential order.

**GETTING THE DATASET:** The collected dataset was organized and saved in Microsoft Excel (MS Excel). The labeled dataset, which was collected to train the supervised ML algorithms, serves as the ground truth for the predictive models being developed.

**IMPORTING LIBRARIES:** This research study imported Python libraries including pandas, numpy, sklearn, plotly, matplotlib and warnings in the source code to preprocess the data sample.

**IMPORTING DATASET:** In the next phase, the dataset was uploaded in the default directory of Jupyter notebook and then imported in the Python source code by means of read_excel() function of pandas library and store it in a DataFrame. The imported dataset is shown in Table 4 and its possible values shown in Table 5.

**FINDING MISSING DATA:** Most of the records were accurate and complete because the data was acquired from existing hospitals. There was, nevertheless, a minor proportion of missing data (or partially completed records), which is 20% records of 950 dataset as shown in Table 2. Thus, records with null values were omitted.

**DATASET EXPLORATION:** Based on the 'Prediction' column in the DataFrame, the dataset is explicitly divided into two binary classes *i.e.*, TB Suspected records (48%) and Not Suspected records (52%). Furthermore, the dataset features are visualized by means of a histogram for each column in the DataFrame using Matplotlib library, as presented in Fig. 4. Discrete labeling was selected to analyze the patterns and correlation among the data features in an easier manner. The labels were derived according to the standard medical thresholds and expert consultations. The labeling approach of imported dataset and possible values shown in Table 5.

**ENCODING CATEGORICAL DATA TO NUMERICAL DATA:** Since ML models work on numerical data, and empirically, our dataset consisted of categorical

---

data values, it is mandatory to encode the categorical columns to numeric columns for better data visualization and machine input. Using the python astype ("category") method, the selected columns in DataFrame were converted to the 'category' data type. Then, by means of the cat.codes attribute, the categorical values were converted to numeric codes. This process is repeated for each categorical column index specified in the dataset.

**CORRELATION MATRIX:** Correlation is a statistical measure that computes the linear relationship or degree of dependence between two variables. In a multivariable dataset, the correlation analyzes the association among variables and organizes them in an applicable matrix data structure called correlation matrix. The correlation coefficient can range from −1 to 1. When correlation coefficient is 1, the variables are positively correlated; when it is −1, the variables are negatively correlated; and when it is 0, the variables are not correlated. Correlation explains the predictive relationship between response and predictor variables. The predictor variables might be the considered features for training the models if there is a significant positive or negative association. To determine multi co-linearity, it exhibits the linear connection between predictor variables. While training, if correlation between predictor variables is more than 0.7 or less than −0.7, one of these variables can be omitted. Only numerical values can be used to determine the correlation coefficient. Python's default correlation method *i.e.*, the Pearson technique is implemented to determine the correlation coefficient in this study. The correlation matrix is demonstrated by means of correlation heat map as shown in Fig. 5. The correlation matrix indicates a significant correlation between cough type and cough length (0.73), as well as cough and cough duration (0.68), reflecting the multicollinearity. It also reveals moderate correlations between ESR value and mucus (−0.55) and between pulmonary effusion and body temperature (0.3). In addition, there is a negligible correlation among crowding, smoking, and exposure. To streamline the model, multicollinearity was dealt with by emphasizing moderate correlations.

**DEPENDENT AND INDEPENDENT VARIABLES:** Using Python iloc indexer in pandas, the feature (independent, explanatory, or predictive) variables and the target (dependent, response, or outcome) variable were extracted from the DataFrame.

**FEATURE SCALING:** Since the dataset features had distinct ranges, the StandardScaler from the python scikit-learn library standardized the feature variables to minimize data anomalies and handle integrity issues. It transforms the numeric column values in the dataset to a common scale without highlighting value range gaps.

**TRAIN AND TEST DATASET:** Research study made use of train-test split to evaluate ML algorithms for predictive solutions. The train dataset, with known statistics, validates the model, while the test dataset makes predictions. Using the train_test_split function from scikit-learn, the dataset was split, with the test dataset comprising 30% and the train dataset comprising 70% of the original dataset.

**Table 4 Imported dataset (760 rows × 18 columns).**

| Patient ID | Cough | Cough duration | Cough type | Mucus | Chest pain | Breathe state | Body temperature | Chills | Pulmonary effusion | ESR value | Diet | Physique | Energy adequacy | Smoking | Crowding | Exposed | Prediction |
|---|---|---|---|---|---|---|---|---|---|---|---|---|---|---|---|---|---|
| 1 | No | 0 | No | No | Yes | Normal | Normal | No | No | Normal | Malnutrition | Weight loss | Fatigue | Yes | Yes | Yes | Not suspected |
| 2 | No | 0 | No | No | Normal | Normal | Normal | No | No | Disturbed | Malnutrition | Weight loss | Fatigue | No | Yes | Yes | Not suspected |
| 3 | No | 0 | Non productive | No | Yes | Normal | Normal | No | No | Disturbed | Malnutrition | Healthy | Fit | Yes | Yes | No | Not suspected |
| 4 | No | 0 | No | No | Yes | Dyspnea | Normal | No | No | Normal | Balanced | Weight loss | Fit | Yes | Yes | No | Not suspected |
| 5 | No | 0 | No | No | Normal | Normal | High | No | No | Normal | Malnutrition | Weight loss | Fatigue | No | Yes | Yes | Not suspected |
| ... | ... | ... | ... | ... | ... | ... | ... | ... | ... | ... | ... | ... | ... | ... | ... | ... | ... |
| 756 | Yes | 0 | Productive | No | Yes | Dyspnea | High | No | No | Disturbed | Balanced | Healthy | Fit | Yes | No | No | TB suspected |
| 757 | Yes | 3 | Productive | Bloody | Normal | Normal | High | No | No | Normal | Malnutrition | Healthy | Fit | Yes | Yes | No | TB suspected |
| 758 | Yes | 3 | Productive | Bloody | Yes | Dyspnea | Normal | No | No | Disturbed | Balanced | Healthy | Fit | Yes | Yes | No | TB suspected |
| 759 | Yes | 3 | Productive | Bloody | Yes | Normal | Normal | No | No | Disturbed | Malnutrition | Healthy | Fit | Yes | No | Yes | TB suspected |
| 760 | Yes | 3 | Productive | Bloody | Yes | Dyspnea | High | No | No | Disturbed | Malnutrition | Weight loss | Fatigue | Yes | Yes | No | TB suspected |

**Table 5  Description of dataset features.**

| Features | Possible values |
|---|---|
| Cough | {0 = No, 1 = Yes} |
| Cough duration | {0 = No cough, 0.5 = Mild, 1 = Occasional, 1.5 = Intermittent, 2 = Moderate, 2.5 = Frequent, 3 = Persistent} |
| Cough type | {0 = No, 1 = Non-Productive, 2 = Productive} |
| Mucus | {0 = Bloody, 1 = Clear, 2 = No} |
| Chest pain | {0 = No, 1 = Normal, 2 = Yes} |
| Breathe state | {0 = Dyspnea, 1 = Normal} |
| Body temperature | {0 = High, 1 = Normal} |
| Chills | {0 = No, 1 = Yes} |
| Pulmonary effusion | {0 = No, 1 = Yes} |
| ESR value | {0 = Distributed, 1 = Normal} |
| Diet | {0 = Balanced, 1 = Malnutrition} |
| Physique | {0 = Healthy, 1 = Weight Loss} |
| Energy adequacy | {0 = Fatigue, 1 = Fit} |
| Smoking | {0 = No, 1 = Yes} |
| Crowding | {0 = No, 1 = Yes} |
| Exposed | {0 = No, 1 = Yes} |

## RESULTS AND DISCUSSION

This section provides the performance metrics, results and discussion of the generated ML-TB predictors. The present research adopted five different supervised ML algorithms including: Decision tree (DT) is a top-down hierarchical structure consisting of three nodes *i.e.*, root (top-most) node, internal (non-leaf) nodes, and terminal (leaf) nodes. Each internal node tests a condition, each branch indicates the test outcome and each terminal node represents class label. It resolves classification and regression problems. It handles both numerical and categorical data. Gaussian naïve Bayes is a type of naïve Bayes, which considers that the features follow a normal distribution, thus making it appropriate for continuous data. Logistic (non-linear) regression predicts the categorical dependent variable using independent variables. It is based on maximum likelihood estimate method. It can be binary, multinomial and ordinal.

Neural networks (NNs) or Artificial NNs are defined as layered network of artificial neurons (nodes) with three major components including learning rules, network architecture (one input layer, one output layer, and one or more hidden layers), and activation function (such as linear, sigmoid, hyperbolic tangent). NNs take the sum of all highly weighted neuron's inputs with bias, process it through an activation function and generate corresponding output. AdaBoost (Adaptive Boosting) is an ensemble that combines numerous weak classifiers into a single strong classifier. AdaBoost was the first successful binary classification boosting technique to be devised. Due to the appropriateness for classification tasks and prior performance in medical diagnostics, these five ML algorithms were chosen for the current research study. More specifically, DT was picked for its interpretability, GNB for its minimalism and efficacy with small datasets,

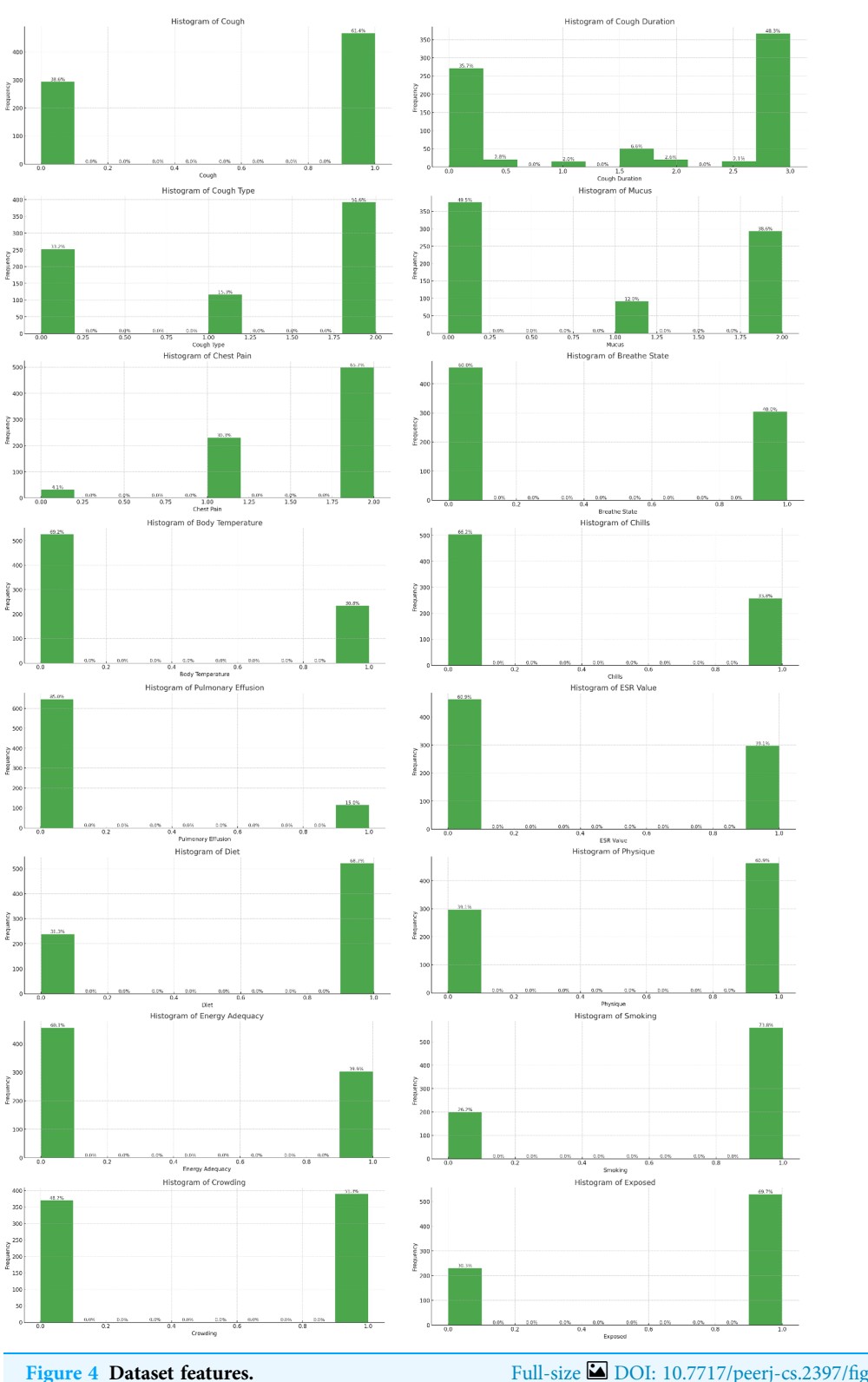

**Figure 4  Dataset features.**

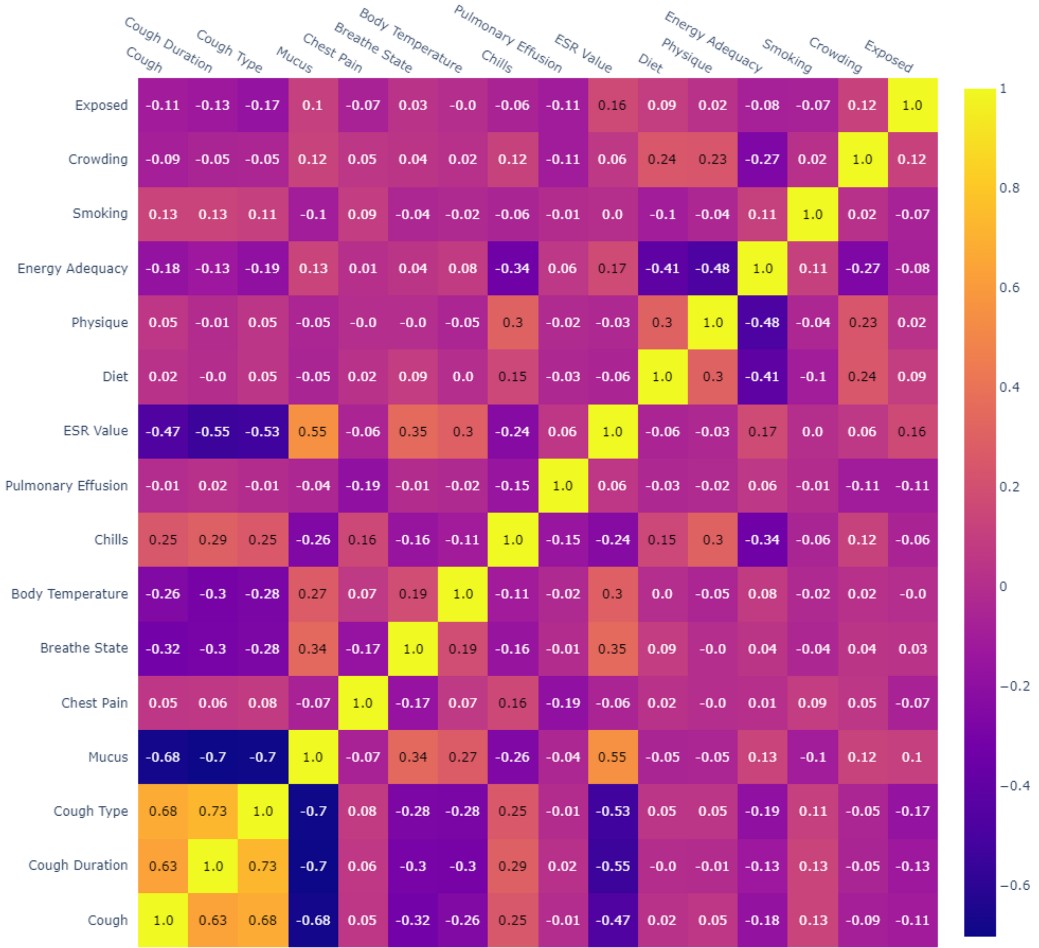

**Figure 5** Correlation matrix.

LRC for its simplicity in binary classification, AdaBoost for its ability to manipulate imbalanced datasets, and NN for its fitness to model complex patterns. For the performance analysis of each classifier, research made use of five different metrics including:

Accuracy defines the fraction of predictions that ML model correctly identifies. Mathematically,

$$Accuracy = \frac{TP + TN}{TP + TN + FP + FN}$$

where TP stands for true positive, TN stands for true negative, FP stands for false positive and FN stands for false negative.

Precision defines the fraction of positive predictions that ML model correctly identifies. Mathematically,

$$Precision = \frac{TP}{TP + FP}$$

**Table 6 Confusion matrix.**

| Actual values | Predicted values | |
|---|---|---|
| | **0 (False)** | **1 (True)** |
| 0 (False) | True Negative (TN) | False Positive (FP) |
| 1 (True) | False Negative (FN) | True Positive (TP) |

**Table 7 Confusion matrix.**

**(a) DT classifier**

| | 0 | 1 |
|---|---|---|
| **0** | 94 | 11 |
| **1** | 7 | 116 |

**(b) GNB classifier**

| | 0 | 1 |
|---|---|---|
| **0** | 92 | 13 |
| **1** | 12 | 111 |

**(c) AdaBoost classifier**

| | 0 | 1 |
|---|---|---|
| **0** | 100 | 5 |
| **1** | 10 | 113 |

**(d) NN classifier**

| | 0 | 1 |
|---|---|---|
| **0** | 96 | 9 |
| **1** | 7 | 116 |

**(e) LRC classifier**

| | 0 | 1 |
|---|---|---|
| **0** | 94 | 11 |
| **1** | 11 | 112 |

where TP stands for true positive and FP stands for false positive.

Recall (sensitivity or true positive rate) defines the fraction of actual positives that ML model correctly identifies. Mathematically,

$$Sensitivity = \frac{TP}{TP + FN}$$

where TP stands for true positive and FN stands for false negative.

F1 score (F-measure or F score) defines the harmonic mean of precision and recall values. It best case is 1 and the worst case is 0. Mathematically,

$$F_1 = 2 \times \frac{precision * recall}{precision + recall}$$

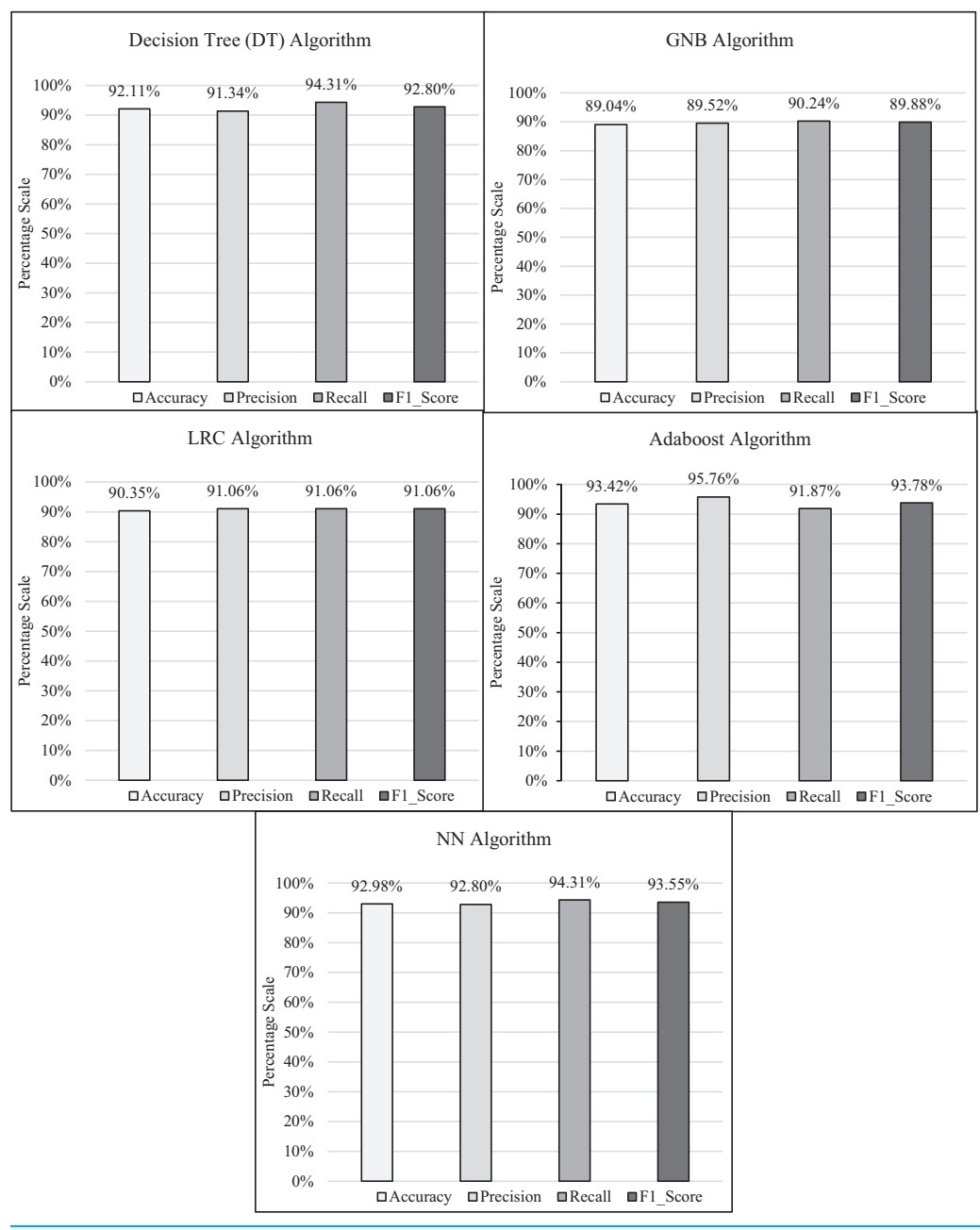

**Figure 6 Performance analysis of individual ML-TB predictor models.**

Confusion matrix is a $N \times N$ matrix that calculates the performance of a classification model, where N represents the number of target classes. It provides a comprehensive picture of ML model performance accuracy and the errors it produces by comparing the actual target values to the predicted values shown in Tables 6 and 7. Table 7 illustrates the confusion matrices for each classifier's performance in diagnosing instances as TB positive (1) or TB negative (0).

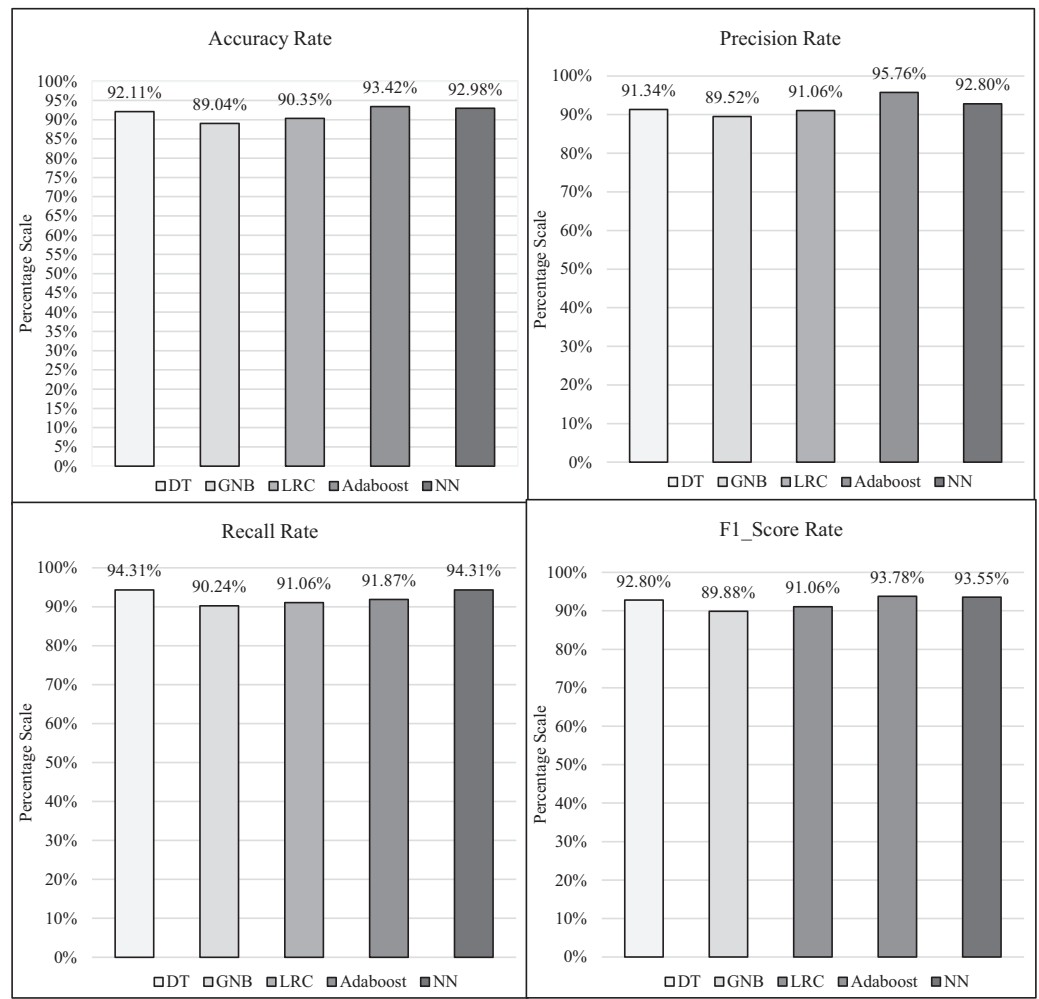

**Figure 7 Comparative analysis of ML-TB predictor models based on performance metrics.**

The research experimental procedures were carried out by means of Jupyter Notebook (Python) where the ML-TB predictor models were trained using a 70-30 train-test split, hyper parameters were regulated *via* grid search and cross-validation was applied to ensure the robustness. The individual as well as comparative performance of each classifier is interpreted as shown in Figs. 6 and 7, respectively. Statistically, the implemented classifiers including DT, GNB, LRC, AdaBoost and NN obtained 92.11%, 89.04%, 90.35%, 93.42% and 92.98% accuracy rate, respectively; 91.34%, 89.52%, 91.06, 95.76% and 92.8% precision rate, respectively; 94.31%, 90.24%, 91.06, 91.87% and 94.31% recall rate, respectively; 92.8%, 89.88%, 91.06%, 93.78% and 93.55% F1 score rate, respectively. The outcomes comprehend that each implemented algorithm successfully anticipated the TB disease. However, the statistical results exposed substantial variances between classifiers, with $p$-values $< 0.05$ for accuracy and precision metrics. Confidence intervals were calculated to measure the robustness of the outcomes. Feature importance analysis including Gini, permutation and SHAP values exposed that the symptoms including persistent cough and

weight loss were most persuasive in predicting TB. AdaBoost ensemble performed far better than other classifiers according to the accuracy, precision and F1 score performance metric. However, sensitivity (recall) is prioritized due to its significance in classifying as many cases as possible. Thus, the DT and NN classifiers are considered as better approach in terms of recall performance metric.

## CONCLUSIONS

In a nutshell, this research study signifies a substantial breakthrough in TB diagnosis, tailored to the exclusive context of Pakistan, particularly in Sindh, where the prevalence of TB is exceptionally high. The present research addressed an essential gap in automated TB diagnosis by emphasizing on context-specific needs and acquiring vital dataset from three notable hospitals in Sindh: ICD Kotri, LUMHS Jamhoro, and Civil Hospital Hyderabad. The in-depth analysis of the study, featuring a variety of ML algorithms such as DT, GNB, LRC, AdaBoost, and NN, yielded positive outcomes. While AdaBoost attained high accuracy (93.42%), precision (95.76%), and F1 score (93.78%), DT and NN classifiers outperformed in recall (94.31%). This study not only optimizes the TB detection with cutting-edge ML algorithms, but it also promotes context-dependent research in healthcare. Despite constraints such as a limited feature dataset and the absence of certain evaluation parameters like as ROC curve (AUC) or calibration metrics, the research study intends to address these in future ventures by augmenting the dataset and leveraging aML tools like WEKA. In addition, the research study expects to widen the scope to incorporate Extra-pulmonary TB manifestations, improve the model with Reinforcement Learning, and develop a ML-TB diagnostic apparatus for seamless integration into the medical setting. By transforming research implications into tangible implements, the research study drives to positively influence healthcare provision and contribute to enriched patient outcomes.

Integrating ML into TB diagnostics can transform early detection and cure, leading to good health and well-being. By implementing accurate ML models, healthcare systems can boost diagnostic precision. Pilot software package in high-burden regions will aid to upgrade these implements, whereas an inclusive training for healthcare providers guarantees operational use and data management. For successful implementation, forming robust monitoring frameworks, refining data collection and privacy benchmarks is crucial. Public awareness campaigns and patient engagement will further drive acceptance, while constant research and global associations will guarantee unceasing novelty. Thus, implementing such policy would positively reinforce the fight against TB, ultimately falling TB mortality ratio.

### Funding

The authors received no funding for this work.

## Competing Interests

The authors declare that they have no competing interests.

## Author Contributions

- Priyanka Karmani conceived and designed the experiments, performed the experiments, analyzed the data, performed the computation work, prepared figures and/or tables, authored or reviewed drafts of the article, and approved the final draft.
- Aftab Ahmed Chandio conceived and designed the experiments, performed the experiments, analyzed the data, performed the computation work, prepared figures and/or tables, authored or reviewed drafts of the article, and approved the final draft.
- Imtiaz Ali Korejo performed the experiments, analyzed the data, authored or reviewed drafts of the article, and approved the final draft.
- Oluwarotimi Williams Samuel analyzed the data, authored or reviewed drafts of the article, and approved the final draft.
- Majed Aborokbah analyzed the data, authored or reviewed drafts of the article, and approved the final draft.

## Ethics

The following information was supplied relating to ethical approvals (*i.e.*, approving body and any reference numbers):

University of Sindh Jamshoro, office of research innovation & commercialization (ORIC) granted ethical approval (from Institutional BioEthics Committee-IBC) to carry out the study within its facilities (Ethical Ref: Letter No. ORIC/SU/311).

## Data Availability

The raw dataset and code are available in the Supplemental Files.

## Supplemental Information

Supplemental information for this article can be found online at http://dx.doi.org/10.7717/peerj-cs.2397#supplemental-information.

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
