# Peer review of "Machine learning based tuberculosis (ML-TB) health predictor model: early TB health disease prediction with ML models for prevention in developing countries"

_PeerJ Computer Science, doi:10.7717/peerj-cs.2397_

## Round 0.1 · original submission · Major Revisions

Thank you for submitting the manuscript to PeerJ Computer Science. Based on the comments from three reviewers, we would like to invite you to revise the manuscript. The major issues include the writing and the report of the parameters the authors used in the manuscript. More details about the methods might need to be addressed for the reproductive purpose. Please let us know if you have any questions in the meantime.

**Language Note:** The review process has identified that the English language must be improved. PeerJ can provide language editing services - please contact us at [email protected] for pricing (be sure to provide your manuscript number and title). Alternatively, you should make your own arrangements to improve the language quality and provide details in your response letter. – PeerJ Staff

Reviewer 1 ·

Basic reporting

1. The authors used clear and unambiguous language in writing the manuscript.
2. There are minor grammatical errors and awkward phrasings throughout the manuscript. A thorough proofreading or professional editing service could help improve the overall quality of writing.
3. The references section is comprehensive but should be updated to include more recent studies, particularly from the last two to three years
4. For section 6 Data Analysis, the steps for data processing are redundant and should either be omitted or moved to the Appendix.
5. In section 7 Results and Discussion line 268 to line 277, the authors described the tables in detail; this also seem redundant, and the information is already presented in the tables.
6. Figure 4, the pie chart, does not add much information and should be omitted.

Experimental design

1. The experimental design is appropriate, and the aims and scope aligns with those of the journal.
2. The research question is well-defined.
3. The authors briefly listed the five models used in the paper. However, the reason for choosing these models as well as a brief description of the models is lacking. I recommend the author add these to the paper.
4. In the Results section, it is recommended to discuss the statistical significance of the results. Include any hypothesis tests or confidence intervals that support the robustness of the findings; that authors could also discuss the importance of different features used in the models. Highlight which features had the most significant impact on model performance and why they are important for TB diagnosis.

Validity of the findings

1. If applicable, provide policy recommendations based on the study's findings. Suggest how health policies could be adapted to incorporate ML-based TB diagnosis tools and improve public health outcomes.

Additional comments

NA

Reviewer 2 ·

Basic reporting

Karmani & Chandio et al. proposed using machine learning (ML) approaches to predict early Tuberculosis (TB) diagnosis. Using ~1,200 self-administered survey distributed among three hospitals in Pakistan, the authors collected responses from patients and employed different machine learning approaches to train classifiers for TB diagnosis. This work is particularly important for TB prevention and diagnosis in developing countries such as Pakistan with high TB burden. The authors provided a comprehensive overview of TB disease and literature review on TB classifiers. Overall, I recommend the authors provide more details in describing features included in the model and model training.

Experimental design

1. Can the authors provide brief description on features included in the survey/model?
2. Can the authors provide more details in describing model training such as hyperparameter tuning?
3. From line 197-199, please provide numerical values or create tables to show percentage of missingness.
4. For figure 5, it would be more informative to plot proportions instead of raw counts for categorical features.

Validity of the findings

1. If possible, can the authors add feature importance to interpret which feature(s) drive this high accuracy of prediction? This will be very informative for epidemiologist to understand the risk factor for TB and thus better manage TB in population.
2. Since different ML models gave different performances on four metrics, can the authors provide insights on which metric is more important in TB early diagnosis for the purpose of model selection? For example, early diagnosis is very important for TB so that high sensitivity is favored over precision.

Reviewer 3 ·

Basic reporting

NA

Experimental design

NA

Validity of the findings

NA

Additional comments

See attached PDF

Annotated reviews are not available for download in order to protect the identity of reviewers who chose to remain anonymous.

---

## Round 0.2 · accepted · Accept

Thank you for submitting the article to PeerJ Computer Science. All reviewers have no further suggestions and agree to accept the article for publication.

Reviewer 1 ·

Basic reporting

I appreciate the authors for revising the manuscript, and writing a clear and detailed response letter. The revision has addressed my previous comments.

Experimental design

No comments

Validity of the findings

No comments

Additional comments

No comments

Reviewer 2 ·

Basic reporting

No more comments

Experimental design

NA

Validity of the findings

NA